# Model Proposal for Service Quality Assessment of Higher Education: Evidence from a Developing Country

Alejandro Valencia-Arias [1,2,*], Claudio Cartagena Rendón [3], Lucia Palacios-Moya [4], Martha Benjumea-Arias [2], Julia Beatriz Pelaez Cavero [5], Gustavo Moreno-López [6] and Ada Lucia Gallegos-Ruiz [7]

1 Escuela de Ingeniería Industrial, Universidad Señor de Sipán, Chiclayo 14001, Peru
2 Facultad de Ciências Económicas y Administrativas, Instituto Tecnológico Metropolitano, Medellín 050005, Colombia
3 Facultad de Administración, Fundación Universitaria Católica del Norte, Medellín 050012, Colombia
4 Centro de Investigaciones Escolme, Institución Universitaria Escolme, Medellín 050005, Colombia
5 Escuela Profesional de Artes & Diseño Gráfico Empresarial, Universidad Señor de Sipán, Chiclayo 14001, Peru
6 Grupo de Investigación en Educación y Ciencias Sociales y Humanas, Institución Universitaria Marco Fidel Suárez, Bello 051050, Colombia
7 Centro de Investigación y Estudios de la Mujer, Universidad Ricardo Palma, Santiago de Surco 15039, Peru
* Correspondence: valenciajho@crece.uss.edu.pe

**Abstract:** Higher education institutions must generate added value through the continuous improvement of services offered to the academic community. Students' needs and expectations must be met to increase their satisfaction within the system. Bearing this in mind, this paper proposes a service quality assessment model for higher education institutions in developing countries. In total, 845 questionnaires were self-administered by university students. The instrument was composed of 119 closed questions. Confirmatory factor analysis (CFA) was applied to create the proposed model. An 18-component model resulted from the data analysis, with an emphasis on academic aspects, infrastructure, web services, wellbeing, and financial procedures. It is expected that higher education institutions in other developing countries may validate, replicate, and adapt this model to their needs.

**Keywords:** higher education institutions; students' perception; service quality assessment; developing countries

## 1. Introduction

Service quality assessment is a challenge for satisfaction models due to the fact that, despite the existence of several factors to measure goods, it is difficult to find a standard measure for services [1]. In addition, service quality is typically harder to assess than goods quality because its temporal reach is broader [2].

### 1.1. Prior Research

There are multiple definitions of higher education quality. It could be defined as the difference between expectations and students' perceptions [1]. This definition considers that, at present, higher education faces the pressure and obligation to add value to student activities, and current trends point toward developing educational value through the continuous improvement of services, increasing student satisfaction [2]. This means that higher education quality can be determined by the degree to which students' needs and expectations can be met [3].

Service quality in the field of higher education is fundamental given that student satisfaction is significantly affected by its positive perception [4]. A clear relationship between educational service quality and student satisfaction and loyalty is thus established [5].

*1.2. Research Problem*

In the education sector, competitiveness is increasing, so it is necessary to continuously monitor the needs and perceptions of the quality of current and potential clients, as well as customer satisfaction (students) [6].

These particularities have led universities to improve their academic quality and services offered to their students [7], thereby achieving quality accreditation from official bodies that consider certification requirements, which relates to students' perceptions of certain elements of the quality of education, such as institutional management and teaching [8]. Given this, it is necessary to have scales and models that contribute to the process of measuring students' perceptions about the quality of services provided by higher education institutions [9].

Bearing in mind the importance of service quality assessment in higher education institutions, as they are the fabricators of new professionals and entrepreneurs, this investigation sets out to propose a model for measuring service quality within these institutions for a developing country in the city of Medellin, Colombia.

## 2. Background

The service sector plays a key role in today's knowledge economy. Service quality evaluation has become a latent need in an increasingly globalized world with the expansion of educational offerings. Therefore, it has become a recurrent topic of interest for researchers in this century [10]. As mentioned in the study of [11], Crosby 1979 defined quality in education as "the conformity of educational outcomes to planned goals, specifications and requirements," so the quality of education has become one of the fundamental objectives of higher education institutions.

The service quality approach is used in higher education to analyze students' perceptions of quality since, as a result of service marketing, the customer is the main contributor in determining the long-term sustainable position of an organization [12]. An emerging strategy to improve service quality in higher education is a student-centered approach, so universities should try to provide the best educational services for students to target student satisfaction and loyalty [13]. The literature has established that the quality of higher education services influences student satisfaction and, in turn, influences institution's image and student loyalty [4].

With the application of technology in education in recent years, educational quality has also been evaluated from an e-learning perspective [13], as well as considering the effects of COVID-19 on the online learning of students in higher education institutions [14]. Educational quality has also been considered in emerging countries [10]. Furthermore, various frameworks and instruments have been proposed to assess service quality in higher education [15].

In the study of [11], perceptions of the quality of educational services, as seen by the students and staff of a higher education institute, were analyzed using the SERVQUAL instrument. The adapted SERVQUAL model can also be used in the context of higher education and identify service quality gaps based on its application in a higher education institution [10]. Given the applicability of the SERVQUAL model, several flexible and applicable models have been proposed in many higher education institutions in developed countries and emerging economies [16].

Three dimensions of quality (design quality, conformance quality, and performance quality) have been considered. However, as mentioned by [17], most quality management initiatives, especially within service industries, fail to be successful because higher education organizations need to measure outcomes. Traditionally, the models proposed to assess educational quality have been based on academic rather than management issues. Therefore, models have been designed in recent years that include a whole set of university management to assess educational quality [16]. This is based on an internal scope to focus on the customer, i.e., the students. At the same time, external evaluations are developed in

the direction of governmental bodies, based on standardized evaluations where the opinion of students can also be considered, although their central focus is more on academic issues.

Nowadays, higher education institutions compete through competitive advantages and high-standard services. Service quality assessment becomes imperative in order to provide information regarding the efficiency of study plans and improvement programs [18]. There is a worldwide effort to achieve student satisfaction and loyalty through the continuous search for quality. Expectations must be met in order to retain students [5].

*Service Quality Assessment of Higher Education*

Service quality assessments offered by different higher education institutions become fundamental in order to guide directors in the design of appropriate programs that may promote, develop, and maintain long-term relationships with current and past students. Student loyalty is gained once a strong relationship has been established through the educational services offered. Ultimately, a loyal community provides financial support for future academic endeavors. In addition, it creates a brand image for the institution, which endures within and without campus life [1].

Preserving and improving service quality is a fundamental requirement for colleges and higher education institutions given the current dynamic and competitive panorama. Globalization; increases in higher education centers, as well as private colleges; and reductions in state financial resources dedicated to public education are all reasons why bettering the offer of education is imperative. Just as private companies have obligations to their clients, universities must answer student demands [15].

Regular assessment is crucial for the management and improvement of the service quality offered within universities. A valid and reliable instrument becomes indispensable in this sense [15]. A great number of models have been developed in attempts to properly measure students' perceptions regarding service quality. Some of them are enunciated below.

SERVQUAL: This is the most well-known and used measure of service quality. Created by Parasuraman, Zeithaml, and Berry in 1985, this scale and some variations of it are used in different service sectors such as banks, sales departments, health, education, and others [19]. The authors describe service quality as a kind of attitude related, yet not equivalent to, satisfaction, and the result of a comparison between expectations and witnessed performance [20]. Adapted to the context of higher education, it must be understood as the difference between what a student expects to receive versus his/her perception of what is actually obtained [19].

SERVPERF: Cronin and Taylor introduced this scale in 1992. The authors criticized the SERVQUAL scale because of its limited validity in different industries as well as cultures [5]. It is argued that its psychometric properties are better [11], and it is even capable of predicting individual answers relating to service quality, provided with better accuracy than the SERVQUAL model [21]. This instrument is focused on the performance levels of various attributes, so service quality is conceptualized in terms of attitudes rather than by the confirmation/non-confirmation paradigm used in SERVQUAL [2].

HEdPERF: This is another scale created to assess service quality in higher education. It did not achieve much popularity due to its similarity to the aforementioned SERVPERF model [5]. The model includes five dimensions: nonacademic aspects, academic aspects, access, program problems, and reputation [22].

5Q'S MODEL: Proposed by [23] to measure the quality of service in higher education institutions. It is composed of five quality dimensions: (a) education and/or research, (b) processes or the implementation of educational activities, (c) infrastructure, (d) communication, and (e) atmosphere. As [18] explains, this model includes factors that can be controlled by the institution and factors that are not explicitly present in the adaptation of the SERVQUAL instrument.

Based on the service quality measurement models, the most representative aspects of each model (in terms of the constructs) have been taken as input for the construction of a questionnaire that is expanded and adapted for higher education institutions. In this

context, variables and constructs contained in SERVQUAL, SERVPERF, HEdPERF, and 5Q'S MODEL have been taken and fused, giving rise to a new model based on an exploratory factor analysis of the reported data.

## 3. Research Methodology

### 3.1. General Background

This paper describes a cross-section diagnostic and descriptive investigation. It has been designed as a quantitative approach based on data collection through self-administered questionnaires. The items inquired about the perceptions and experiences of students regarding obtained services and attention within different areas of the educational institution. The respondent sample was composed of students enrolled in on-campus courses. The questionnaire was thus answered within the institution's facilities.

### 3.2. Tools and Procedures

This type of methodological design was used because the purpose was to collect information from the primary source: current students. The self-administered questionnaire was composed of 119 closed questions fashioned into a Likert scale.

A pilot test was conducted to evaluate the instrument and population reactions, test variables, and estimate the time required to gather data. Interviewers were trained in the application of said instrument so as to unify the procedures to be applied during the formal data collection. The sample for the pilot test was composed of 20 enrolled students.

The 119 questions used in the questionnaire were taken and adapted from other models to measure the quality of service proposed in the theoretical framework (SERVQUAL, SERVPERF, HEdPERF, and 5Q'S MODEL), but we sought not to bias their relationships, allowing us to obtain the form in which the questions were grouped based on confirmatory factor analysis.

### 3.3. Sample

Inclusion criteria for sampling included enrollment in on-campus courses, as well as participation during different academic days programmed by the institution. In total, 845 college and active students from the city of Medellin were selected using the probabilistic technique and subjected to the instrument. Afterward, questionnaires were filtered to fulfill the validity criteria and narrow the number to a total of 814 self-administered questionnaires.

### 3.4. Data Analysis

The 23rd version of the SPSS software was used to process and analyze the data. The database was screened, and a descriptive univariate analysis was run. Subsequently, an exploratory factor analysis (EFA) was performed to obtain a base for the theoretical model to be presented and, further on, to validate it with a confirmatory factor analysis (CFA).

The employed procedure entailed the analysis of each set of variables according to the following: First, an EFA was applied to the whole group of variables using maximum likelihood estimation (MLE) as the method for factor extraction, and VARIMAX was used for rotation. Second, an 8-hypothesis model was proposed according to factors retrieved through the EFA. Lastly, this model was tested through CFA, so standardized factor loads were corroborated for each variable.

## 4. Results

A rotation of axes was initially applied in order to clarify the factor structure. An orthogonal rotation method that minimizes the number of variables according to high saturation on each factor (VARIMAX) was used, and 22 factors were obtained for analysis out of the 119 original variables. Eighteen of those contain two or more items. They are grouped as follows:

**First Component—Welfare University (WU):** This component is associated with variables related to service satisfaction regarding university wellbeing offered by the institution.

The evaluation reported aspects such as scholarship applications and processes, means to communicate welfare services, benefits offered by partnerships with companies, coverage of welfare services by the university, the quality of service offered by university personnel, conditions of recreational and cultural facilities, ways to convey information regarding socioeconomic services as well as the department's general function.

**Second Component—Library Services (LS):** Variables grouped in this factor correspond to those that evaluate student satisfaction with services offered by the library system. This is why each item measures students' perceptions of attention received by library personnel, practicality, and the usefulness of bibliographic databases, as well as technical updates performed on them.

**Third Component—Admissions and Registration (AR):** Variables contained in this component make reference to the satisfaction experienced by students regarding admissions and registry offices. The quality and reliability of the information obtained regarding admission formalities and procedures, counseling for enrollment, amiability, respect, and attention and interest shown by the counselor are all assessed in order to detect opportunities and offer and introduce better practices for these services.

**Fourth Component—Cafeteria and General Services (CGS):** This factor comprises variables pertaining to the measurement of general services provided by universities and their cafeterias. These items aim to identify the celerity with which the personnel operate in terms of solving demands, the range of offers, product quality, and prices. Considering an ever-changing and competitive market, it becomes imperative to answer client demands and create a positive image regarding service in consumers' minds.

**Fifth Component—Physical Infrastructure (PI):** This refers to variables related to buildings, facilities, and infrastructure conjoined to education and training strategies by the university. Specifically, it seeks to measure the proportion of built structures to open areas, the maintenance of physical resources (chairs, windows, boards), study rooms (desks, capacity, lighting, ventilation, etc.), and the cleaning and maintenance of other main areas (e.g., cafeterias, classrooms, and auditoriums), among others.

**Sixth Component—Academic Aspects (AcAs):** This is associated with variables related to satisfaction regarding academic services offered by the institution. The services assessed respond to criteria such as academic requirements; the broadcast of information of interest, i.e., conferences, forums, cultural events, partnerships, and extracurricular activities offered to the student body; teaching methods, such as the understanding of subjects; and professors' services and vocation responsibilities while verifying the fulfillment of study programs.

**Seventh Component—Services Supported on the Web (SSW):** This factor comprises variables designated to measure student satisfaction regarding the Internet and online services. Students' perceptions of the efficacy of online communications via institutional email accounts and Wi-Fi availability in different areas (cafeteria, hallways, classrooms) are estimated.

**Eighth Component—Deanship Services (DS):** This is composed of two items that establish a relationship with satisfaction experienced by students using attention services. Two fundamental axes are considered: On the one hand, attention schedules offered by the Deanery and Academic Vice Chancellery, their response, facility conditions, and the broadcast of information concerning these departments. On the other hand, a timeline is established, as well as interest in solving academic issues on behalf of these two instances.

**Ninth Component—University Extension Services (UES):** This component is associated with variables that pertain to the fulfillment of college extension services offered by the institution. Courses offered for extension programs, announcement timeliness, advertisement tools, easy access to information, and teaching methods used by professors in extension programs are taken into account here.

**Tenth Component—Administrative Staff (AS):** Variables contained in this component correspond to the analysis of students' perceived satisfaction concerning attention offered by administrative personnel. Managers, chiefs of staff, and administrative assis-

tants are reviewed regarding availability and attention quality. Regular channels to submit petitions, complaints, suggestions, and praise are also examined. This is an important factor given the fact that a large percentage of students transfer to other universities when they perceive low-level benefits in this sense. The right combination of listening to students' thoughts and offering them a good service is considered a propeller to success.

**Eleventh Component—Research Center Service RCS:** This factor includes variables related to the fulfillment of the services offered by the institution's investigation center. Communication methods and information available concerning types of degrees, responses to needs, and concerns and suggestions from the study body are hereby addressed.

**Twelfth Component—Computer Facilities (CF):** This conforms to two items related to the satisfaction experienced by students regarding computer rooms. Equipment quality (computer speed and reliability), facilities, space distribution, access to specialized software according to the interest, and the needs of students are measured.

**Thirteenth Component—Physical Plant Adequacy (PPA):** This component evaluates variables pertaining to the adequacy of wired-connection facilities offered by the institution: the proportion of buildings to open areas, spaces designed for student counseling, connectivity, and the speed of wired Internet connections in computers.

**Fourteenth Component—Psychological Services (PS):** This factor includes variables that measure student satisfaction with regard to psychological services offered on campus. The items assess students' perceptions of schedules established for psychological services (set appointments and consultation provided), waiting times, facilities, and the service quality offered by professionals.

**Fifteenth Component—Portfolio Services (PS):** Variables grouped in this factor correspond to those that evaluate student satisfaction with services provided by portfolio systems within the institution. Items designed to explore this address media used to disseminate service information, the service quality offered by portfolio officials, the reliability of the information delivered, the satisfaction levels reported regarding formalities, and acceptance procedures.

**Sixteenth Component—Financial Procedures (FP):** This component is associated with variables pertaining to satisfaction in relation to financial applications made to the institution. The services evaluated include aspects such as attention provided during petitions for payment agreements with the institutions, procedures for financial matriculation and scholarships, timelines for delivering requested documents (certificates, records, diplomas), and payment deadlines.

**Seventeenth Component—Audiovisual Aids (AA):** Variables in this component refer to the satisfaction experienced by students when using audiovisual tools during study sessions and classes. The amount of equipment available; accommodations to reserve these tools and/or spaces; and equipment installed in the space, i.e., TVs, PCs, wired Internet connections, audio equipment, convenient electric sockets, etc. are considered in order to detect opportunities and for institutions to offer and introduce better practices for these services.

**Eighteenth Component Online Answers (OA):** The final component is related to variables regarding response times estimated for demands or requirements established via chats and WhatsApp.

During this first stage of analysis, it was possible to determine that collected data in the investigation do not seem to show redundant information given that factor loads were mostly higher than 0,6 [24]. This allowed the mean of the factor loads in each construct to be higher than 0,7 [25], thus achieving convergence in the model. Following this analysis, Figure 1 shows the model hypothesis proposed for validation.

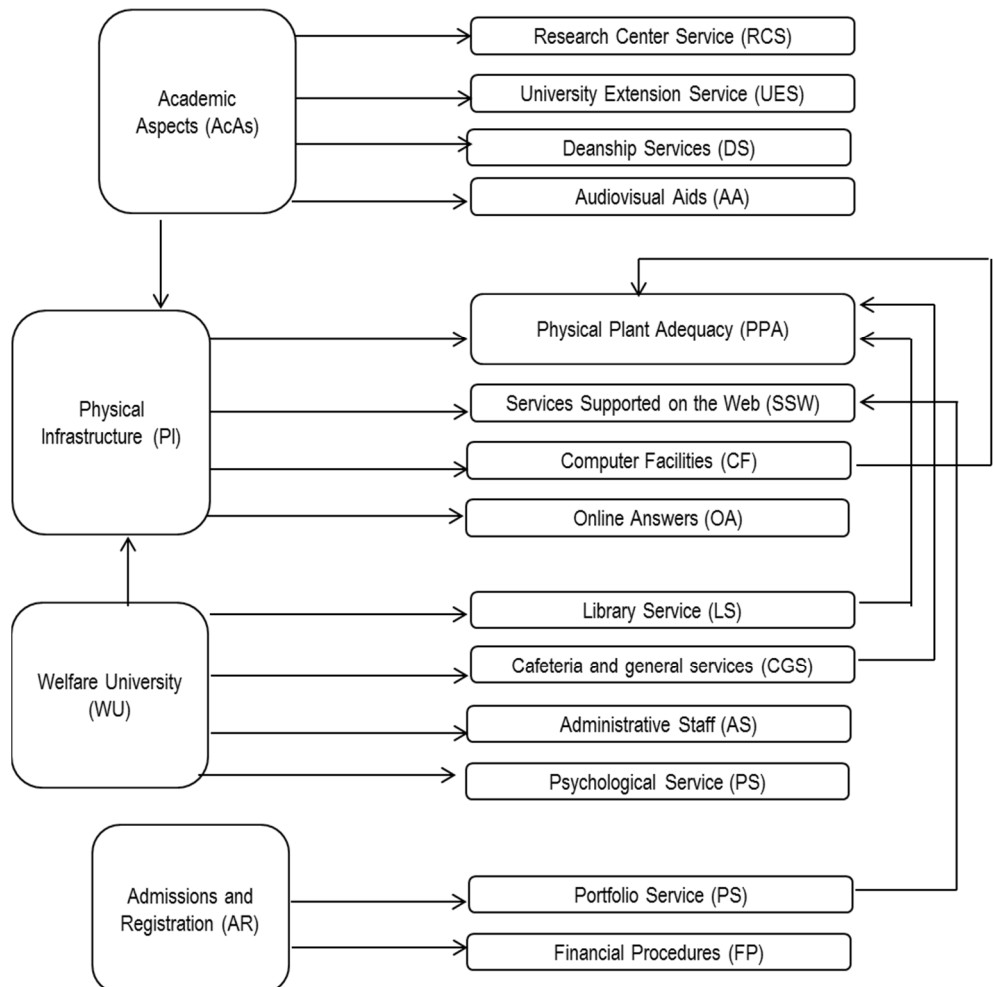

**Figure 1.** Proposed model for assessing levels of satisfaction in students regarding different services offered by the institution. Source: created by the author with SPSS 23.0 statistical software.

The proposed model explores the present relationship between the different variables that have been extracted from SERVQUAL, SERVPERF, HEdPERF, and 5Q'S MODEL; therefore, the associations that may exist between each variable were checked, and the constructs in which they can be grouped were reviewed. In addition, it is important to mention that, for all the included variables, the level of satisfaction of the students in terms of each evaluated aspect is measured, as is the reason why each factor implies a component of satisfaction. In addition, it is important to clarify that the purpose of the model is to identify the relationships and influences between the variables, which is why we did not explicitly define the variables as independent or dependent; for example, in the case of "computer facilities", the dependent variable is "Physical infrastructure", but it behaves like an independent variable for "Physical Plant Adequacy".

Once the exploratory factor analysis (EFA) was completed, the confirmatory factor analysis (CFA) was run because it is a common practice in obtaining valid evidence for theoretical models obtained through EFA. This method allows researchers to demonstrate the validity of the factor structure previously obtained through EFA and, therefore, the validity of theoretical conclusions arrived at. Accordingly, models obtained through EFA are usually validated with CFA [26]. Thus, the procedure performed consisted of validating scales used to measure information and confirming instrument reliability.

### 4.1. Convergent Validity

As aforementioned, the validity of the measurement scales, each construct, and the whole instrument was tested through the statistical method of confirmatory factor analysis [27]. Bearing this in mind, it must be taken into account that model reliability is weighed on two levels: on the one hand, the reliability of observable items, and on the other hand, construct reliability [28]. The removal of indicators was not necessary during the study we conducted because standardized factor loads met the criteria established by the authors previously cited.

In Table 1 results of Barlett's test and the KMO test are shown. These statistics were chosen since they are the most adequate for sample testing in this model [29]. A p-value must be lower than the critical values of 0.05 or 0.01 because significance is reached if it is higher than 0.05, and it would not be possible to reject the null hypothesis for sphericity. Consequently, this would make it impossible to guarantee that the factor model is the best explanation for the obtained data [30]. Considering that the proposed model presents the results of Bartlett's test to be equal to zero, it can be said that there are significant correlations between the variables.

**Table 1.** Convergence validity for KMO and Bartlett's test. Source: created by the author with SPSS 23.0 statistical software.

| Factor | Value (KMO) | Value (Bartlett) | Fulfill Criteria |
|---|---|---|---|
| Audiovisual Aids (AA) | 0.696 | 0.00 | Yes |
| Physical Plant Adequacy (AII) | 0.618 | 0.00 | Yes |
| Computer Facilities (CF) | 0.824 | 0.00 | Yes |
| Physical Infrastructure (PI) | 0.909 | 0.00 | Yes |
| Online Answers (OA) | 0.500 | 0.00 | Yes |
| Academic Aspects (AcAs) | 0.930 | 0.00 | Yes |
| Admissions and Registration (AR) | 0.938 | 0.00 | Yes |
| Library Services (LS) | 0.919 | 0.00 | Yes |
| Welfare University (WU) | 0.940 | 0.00 | Yes |
| Portfolio Services (PS) | 0.774 | 0.00 | Yes |
| Cafeteria and General Services (CGS) | 0.909 | 0.00 | Yes |
| Research Center Services (RCS) | 0.866 | 0.00 | Yes |
| Deanship Services (DS) | 0.866 | 0.00 | Yes |
| University Extension Services (UES) | 0.885 | 0.00 | Yes |
| Administrative Staff (AS) | 0.858 | 0.00 | Yes |
| Psychological Services (PS) | 0.858 | 0.00 | Yes |
| Services Supported on the Web (SSW) | 0.882 | 0.00 | Yes |
| Financial Procedures (FP) | 0.769 | 0.00 | Yes |

### 4.2. Discriminant Validity

Discriminant validity is one of the most common criteria applied to evaluate scales used for measuring latent constructs in social sciences. It must be stated during this stage that for measurements to be valid, variables must significantly correlate to those that are similar to themselves and that the correlations must be higher than those between the variables and those proposed in a different construct [31].

Discriminant validity in the present investigation was examined by testing the confidence interval for the correlations estimated for each set of factors, which should not be valued at one [32]. This criterion was met in all cases, so it can be said that the instrument is valid since it measures, to a high degree, what was intended.

The instrument's internal consistency and reliability were estimated with Cronbach's alpha because it is a statistical tool that considers items on a Likert scale, measuring the same construct and determining if they are highly correlated [33]. As can be seen in Table 2, the instrument seems to have adequate internal consistency and reliability because all Cronbach's alphas are within the range of values suggested by previously cited authors.

**Table 2.** Reliability index—Cronbach's alpha. Source: created by the author with SPSS 23.0 statistical software.

| Factor | Cronbach's Alpha |
|---|---|
| Audiovisual Aids (AA) | 0.870 |
| Physical Plant Adequacy (AII) | 0.787 |
| Computer Facilities (CF) | 0.925 |
| Physical Infrastructure (PI) | 0.927 |
| Online Answers (OA) | 0.85 |
| Academic Aspects (AcAs) | 0.928 |
| Admissions and Registration (AR) | 0.925 |
| Library Services (LS) | 0.954 |
| Welfare University (WU) | 0.947 |
| Portfolio Services (PS) | 0.886 |
| Cafeteria and General Services (CGS) | 0.929 |
| Research Center Services (RCS) | 0.953 |
| Deanship Services (DS) | 0.928 |
| University Extension Services (UES) | 0.944 |
| Administrative Staff (AS) | 0.894 |
| Psychological Services (PS) | 0.963 |
| Services Supported on the Web (SSW) | 0.889 |
| Financial Procedures (FP) | 0.816 |

It is opportune to say that results from the CFA constitute evidence of the existence of a viable factor analysis model for the identification of factors that impact students' perception of service quality offered by higher education institutions. Convergent and divergent validity within the instrument, together with acceptable reliability, confirms that the instrument measures fundamental variables that have direct and indirect impacts on student experiences in this investigation.

*4.3. Data Analysis and Hypothesis Testing*

An estimation of the structural model proposed was subsequently performed in order to assess the service quality offered by higher education institutions. The hypotheses proposed were tested, as well as the degree of their association, using Somers' D, which measures ordinal associations between two variables taking an absolute value ranging from −1 to 1. Values closer to one indicate a strong relationship between the variables, whereas values closer to zero indicate few to no associations between the two variables [34].

Statistics extracted from the SPSS software were organized via crosstabulation in order for it to become clear which variables were associated with the hypothesis and which were not. The goal was to confirm the association for hypothetical relations and to corroborate that, among the other constructs, there were no high association levels. Figure 2 shows the model for student satisfaction regarding services offered by the institution.

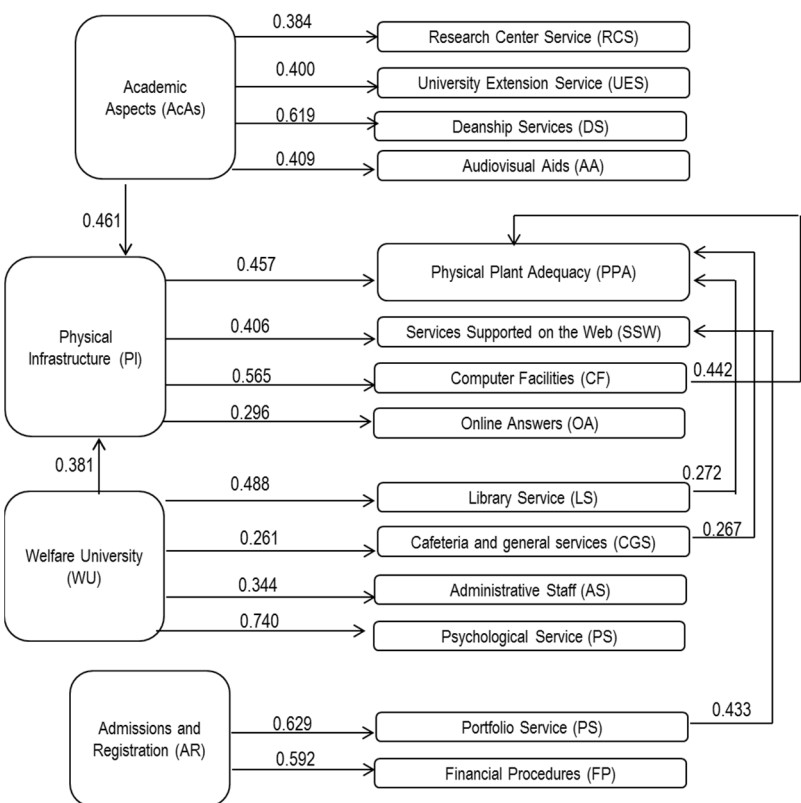

**Figure 2.** Model of student satisfaction regarding services offered by the institution. Somers' D was used for the proposed model. Source: own elaboration. Arrows point to the null hypotheses confirmed in this investigation.

## 5. Discussion

Customer service has become a fundamental process for all types of organizations. Its main objective is to achieve client satisfaction with any of the processes that customers may interact with frequently, regardless of the number of encounters in a set timeframe. Higher education institutions count on several processes that, by a great majority, are used by all its students.

Higher education institutions have two objectives in measuring customer service satisfaction: the first one relies on regulations demanded by national education laws, and the second one is the basis of the institution and, therefore, must always be taken into account. The authors of [35] establish that students are the users who directly receive education, and who better than them to evaluate their experience? Even though their assessment repeatedly relies on academic achievement, their opinion is indispensable and must be considered by the institution in order to create trust in their relationship with the student body. It is important for students to perceive that channels of communication are open throughout the whole process.

Measurement models for satisfaction represent, as stated by [36], measure indexes or variables that can facilitate comparison within an industry—in this case, the higher education sector. This can be useful, as forecasts can be made aiming to implement better practices and create added value for students and the rest of the academic community.

Variables and indexes can also help identify which aspects must be assessed in terms of satisfaction so as to determine which are the most important characteristics. A clear course of action can thus be established for future investments, generating competitiveness to face future challenges in the educational sector both locally and nationally.

It is of critical importance for current organizations to assertively identify the most relevant aspects in relation to customer service. All aspects are vitally important when considering a higher education institution since students interact with all college areas

throughout the duration of their academic careers. Analyzing the importance levels of each variable is a challenge because doing so requires institutions to clarify the investments to be made and any significant needs according to satisfaction assessment results. Results will also construct critical value chains and allow for adaptations to processes and procedures in order to respond to student needs in terms of their satisfaction and comfort in their institution of higher education.

Our model will add to the above because it provides a method to identify moments of truth, not only in the macro-academic process, but also in each of the areas that are part of said process, such as admissions and registration, online platform interactions, approaches to academic directors, and institutional wellbeing, among others. This entails the improvement of the learning experience because it is not about the use of a product; instead, it is about the experience and its image [37].

*Comparison to Other Models*

The proposed model for satisfaction assessment is determined based on the experience of satisfaction with customer service and reinforced with moments of truth (MoT), defined by [38] as "any episode in which the customer comes into contact with the organization and gets an impression of its service". The author adds that numerous moments of truth are the components of product services. Considering this definition, it is established that satisfaction measurements in a higher education institution should acknowledge all areas that comprise customer service since all students interact with all of them at some point throughout their academic lives.

The authors of [35] propose a model for measuring satisfaction that takes 11 different dimensions into account. These variables comprise aspects related to the student as validators of the process in regard to personal results such as achievements, recognition of success, self-realization, etc. Although invaluable, they do not ultimately determine service satisfaction because human beings, in their personalities and behaviors, include different reactions and actions according to individual life processes. These are variables with significant relevance when measuring the impact of education in universities, yet they do not significantly contribute to determining whether or not there is satisfaction with a service.

The authors also understand that there are aspects of service that extend from measurement, such as infrastructure, security, and teaching methods, which end up being fundamental to the product per se. The author of [6] proposed a 10-variable model for a Peruvian college, Ricardo Palma University. The author determined two subjective variables in it: college climate and personal and social attitudes (self-perception). These are variables that depend more on individual perception rather than on the palpable services offered. The author is more objective with dimensions, taking into account measured satisfaction values regarding important common areas, e.g., cafeterias, computer rooms, and laboratories. These variables are closer to what Albertch said about interactions between students and college services.

The authors of [7] proposed a model based on 4 dimensions for the Central University of Venezuela comprising a total of 52 questions. The dimensions are (1) teaching, which relates to all academic activities (classes, evaluations, pedagogy, methodology); (2) academic organization, understood as class programs, schedules, and access to teachers and administrative personnel; (3) infrastructure and college services, which refer to concrete aspects, e.g., facilities, classroom, common areas, security, etc.; and, finally, (4) college life, a dimension including subjective qualities (e.g., skill formation and personality traits) that educational processes cannot totally guarantee because it depends on individual attitudes and cognitive aptitudes.

As can be noted, the previously mentioned models lack integrality in the components of academic products and in the moments of truth analysis that entails academic exercise for a college student. This model seeks to propose a holistic instrument with which the satisfaction of all variables included in the academic product of a university can be analyzed.

Ultimately, it seeks to create a method of assertive and holistic decision-making to add value and improve offered services.

One of the most important aspects when evaluating organizational quality is assessing customer satisfaction. Students, being the main users of a university, are the ones who can best evaluate the quality of educational services, especially since, in the field of education, satisfaction is also understood as the client's perception of how their needs, goals, and desires have been fulfilled [39]. For this reason, relationships are established between the different services in the proposed model, aimed at getting to know the behaviors and influences between one another. Complementarily, the different services are contrasted in the following constructs: academic issues, infrastructure, college wellbeing, admissions, and registry. The results from these hypothetical relationships show that services related to academic issues have a significant correlation with those offered by the Deanery and Vice Chancellery (0.619), revealing that the informative and communicative nucleus between a college and its students is focused on those positions that can create a good experience for the student. This becomes the key element in changing and improving the perception the student body has regarding the institution and its leaders.

Hence, in the future, different student needs and levels of satisfaction can be answered and increased. In a similar fashion, facilities and infrastructure have a strong relationship with the functioning of spaces such as computer rooms (0.565); college wellbeing is highly related to psychology services (0.740); and portfolios and related systems for payment requirements and financial proceedings correlate with high values with services offered by admissions and registry offices (0.629 and 0.595, respectively). This corroborates the importance of contact centers students have access to since they become key strategic assets for institutions and set the course toward differentiation and attention quality.

A study performed in Colombia showed that students' satisfaction is related to their perceptions of academic excellency requirements, career rankings, and the academic process, as this is how they perceive intellectual growth. Furthermore, family climate was found to be a direct influence [18].

Studies conducted in Malaysia revealed that when students are satisfied with received services their perception of the institution's image improves, and this can have a positive effect on their loyalty. Nevertheless, these results are worrying for researchers considering Malaysian universities' bad reputation among international students [4].

Moreover, university extension services, audiovisual aids, buildings, and facilities reveal a high level of association with the academic aspects construct. This may indicate that assisted services not only allow clients to interact with intermediaries and receive personalized attention but also enable the institution to establish a liaison with its users. Building relationships between institutions and students makes it easier for them to benefit from the institution's growth and position through the implementation of dissemination channels regarding bachelors and postgrad programs. It can also inform better and new politics of academic exchanges or internships, as well as any other kinds of information that provides alumni a sense of belonging and sets guarantees for newcomers.

Finally, we argue that student satisfaction is a key element in the estimation of education quality because it reflects the efficiency of academic and administrative services. Satisfaction with learning units, interactions with teachers and classmates, facilities, and infrastructure must all be taken into account. Students' perspectives, based on their experiences, expectations, and needs are undoubtedly a variable that will serve as an indicator for managerial improvement and academic program development.

## 6. Conclusions and Implications

Satisfaction assessment models are strategic tools that serve as starting points for specific interventions and for the design of lines of action. The objective is to create a permanent and continuous improvement process regarding relevant aspects of the provision of services. Eventually, this will allow for a more accurate direction for financial, technological, and human resource investments.

Measuring satisfaction is important for higher education institutions due to the technical and procedural requirements suggested and set by the Ministry of National Education in Colombia. By using an instrument, these requirements can be better evidenced; ergo, institutions will be more accountable and responsible in fulfilling their satisfaction goals. Institutional accreditations and academic programs will follow. Furthermore, it supports all quality areas where evidence is required to know what clients think about said service.

It is important that an instrument for customer service satisfaction properly measures each and every one of the areas comprehended in a higher education institution. Otherwise, at no time will a precise result of satisfaction or dissatisfaction be achieved if the instrument fails to inquire about all forms of interactions between students and the institution. Therefore, assertive decision-making will not be possible and neither will the implementation of development programs seeking required accreditations, thus failing to obtain quality certificates. Specific surveys could be conducted in certain areas, but that would exhaust the students being sampled and damage the results in the long run.

When specific improvement strategies are determined in the academic processes of higher education institutions, efforts are made to improve the procedures that students must carry out so as to streamline processes that involve several service areas. These procedural improvements and satisfaction with service measurements are fundamental stages when establishing an institutional culture and consolidating its image in a market overflowing with academic offerings.

The model proposed in this study builds on existing models based on quality, with applications for a developing country. Therefore, this model for evaluating educational quality could be applied in other regional contexts, such as Latin America. In this sense, universities and institutions of higher education could consider this model to determine the opinions of students on the quality of education. Similarly, the study has limitations in the context of its application since the study population was focused on a single city in Colombia; the model should also be applied to other cities in the country for comparison.

Statistically, model reliability was analyzed with Cronbach's alpha, which determines consistency within the instrument. The indicator is also useful for determining the reliability of variables studied in the questionnaire. The objective was to create an instrument that can be adapted to different higher education institutions. One would just need to add or suppress some areas considering differences in certain services (given that each institution provides different services). This instrument proves to be a helpful tool for future assessment processes regarding student satisfaction with services offered by institutions of higher education.

We observed that among the academic aspects the most influential is Deanship services (DS), which may be due to the fact that they are the dependencies with which students have direct contact in solving their concerns and problems about their curriculum issues, which is the reason why students see significant value in these services. In addition, they consider the University Extension Service (UES) to be important, where language-teaching programs and other courses are included to complement the students' training; in this context, they positively value flexibility being close to the needs of the market. Furthermore, they highlight the need to include new teaching strategies and business sector tools to improve their work skills.

Regarding the aspects most valued in the physical infrastructure construct, Computer Facilities (CF) and Physical Plant Adequacy (PPA) are highlighted. Regarding Computer Facilities (CF), the offer of computer rooms equipped with software and appropriate digital tools for the accomplishment of work and consultations is valued positively. In addition, the connectivity and speed of the wireless network for the Internet are evaluated throughout the campus, which facilitates consultation and access from any device. As for Physical Plant Adequacy (PPA), there is a special impact on the types of facilities and infrastructures that the university integrates into its training strategy in relation to free spaces, the maintenance of resources for class development, and the cleaning and maintenance of recreational areas.

Regarding the Welfare University (WU) component, it highlighted the positive valuations that the surveyed students presented about Psychological Services (PS), which also highlights the relevance of schedules, physical facilities, and speed in the assignment of appointments. In addition, Library Services (LS) is highlighted too due to the need to support the studied topics with autonomous work and different complementary material besides those provided in classes. In addition, the physical space of the library and its endowment with computers and specialized bibliographic databases is highlighted as an essential component of library services.

Finally, as future work, we suggest consolidating the model toward a clear definition of constructs and variables, taking the main variable as satisfaction with the offered university services and including new factors that higher education institutions have been incorporating into their mission, such as those mentioned in [40], given the greater propensity of university students to create high-value-added companies.

The model can be replicated in institutions of a technological nature or with blended programs, but in this case, factors that measure students' acceptance of e-learning tools should include the creation of virtual object learning programs using mobile devices [41], learning with mobile devices [42], the repercussions of adding more and more virtual subjects to the curriculum [43], and the implementation of ICT-based learning communities [44].

Future studies should incorporate new components for quality measurements, suggesting triangulation with other actors in the process, such as teachers with institutional trajectory [45], formative research processes [46], and transfers of knowledge [47], among other emerging factors in university education processes.

Additionally, we suggest using the model again by contrasting the results reported between students from different programs or students from different semesters, differentiating the way in which student satisfaction changes while they remain in higher education institutions.

We suggest validating the model with focus groups of experts in university services and from there more directly identifying the typology of strategies that can be raised from the results so that it becomes a replicable model that is transversal, allowing for the periodic monitoring of satisfaction with university services.

**Author Contributions:** Conceptualization, C.C.R., L.P.-M. and J.B.P.C.; methodology, J.B.P.C. and A.L.G.-R.; software, G.M.-L.; validation, A.V.-A., M.B.-A., L.P.-M. and G.M.-L.; formal analysis, A.V.-A. and C.C.R.; investigation, A.L.G.-R. and L.P.-M.; resources, G.M.-L. and A.L.G.-R.; data curation, G.M.-L. and M.B.-A.; writing—original draft preparation, A.V.-A., L.P.-M. and M.B.-A.; writing—review and editing, A.V.-A., A.L.G.-R., M.B.-A., G.M.-L. and C.C.R.; visualization, C.C.R.; supervision, A.V.-A.; project administration, A.V.-A. and L.P.-M.; funding acquisition, G.M.-L. and A.L.G.-R. All authors have read and agreed to the published version of the manuscript.

**Funding:** This research received no external funding. The APC was funded by Universidad Señor de Sipán.

**Institutional Review Board Statement:** The study was conducted in accordance with the Declaration of Helsinki, and approved by the Ethics Committee of Institución Universitaria Escolme (protocol code 13072020).

**Informed Consent Statement:** Informed consent was obtained from all subjects involved in the study.

**Conflicts of Interest:** The authors declare no conflict of interest.

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
