# Peer review of "Model Proposal for Service Quality Assessment of Higher Education: Evidence from a Developing Country"

_education, doi:10.3390/educsci13010083_

Round 1

Reviewer 1 Report

The literature review is poor. It is recommended to conduct a bibliometric analysis to determine the relevance of the topic.

The title of the article and the introduction should be clarified. The research results are local and it is not entirely clear how they will be applied to developing countries. What region of results implementation is proposed?

In the literature review, it should be noted that there are internal and external assessment of the quality of education. How will the results of the internal quality assessment affect the external quality assessment? This should be reflected in the text of the article.

What practical recommendations as a result of research? The results are obtained and analyzed, but the model for implementing the results is not clear.

Author Response

Dear Prof. Dr. James Albright

Editor-in-Chief

Education sciences

Kind regards

According to the review of our article by the reviewers, the following changes were made, properly in the manuscript:

Reviewer

Comment

Response

Reviewer 1

The literature review is poor. It is recommended to conduct a bibliometric analysis to determine the relevance of the topic

The literature review section is enriched to demonstrate the relevance of the topic

Reviewer 1

The title of the article and the introduction should be clarified. The research results are local, and it is not entirely clear how they will be applied to developing countries. What region of results implementation is proposed?

It is clarified in the title of the article and the introduction that the study focuses on a developing country. The place of study is included

Reviewer 1

In the literature review, it should be noted that there are internal and external assessment of the quality of education. How will the results of the internal quality assessment affect the external quality assessment? This should be reflected in the text of the article.

The information requested in the literature review is included.

Reviewer 1

What practical recommendations as a result of research? The results are obtained and analyzed, but the model for implementing the results is not clear.

Practical recommendations are included in the conclusions and implications section.

Reviewer 2

This is an original study and a nish are to be investigated. The literature review, references could be extended.

The literature review section is enriched to demonstrate the relevance of the topic

Reviewer 2

The research and analysis is fine. The outcome, discussion and recommendation could be improved as well as specific highlights from the study could be presented clearly for further research.

Practical recommendations are included in the conclusions and implications section.

We look forward to your comments and hope to hear from you soon.

Thank you very much

_

The authors

Reviewer 2 Report

This is an original study and a nish are to be investigated. The literature review, references could be extended. 

The research and anlysis is fine. The outcome, discussion and recommendation could be improved as well as specific highligths from the study could be presented clearly for further research.

Author Response

(The authors gave the same response as above.)

Reviewer 3 Report

An excellent example of research and presentation of results: a clear and logical structure, a strong review of the literature, a clearly described methodology, conclusions are logical and based on the study.

Author Response

(The authors gave the same response as above.)

Round 2

Reviewer 1 Report

All is ok. Thanks to authors for their good work.